# The Bidirectional Relationship between Chronic Kidney Disease and Hyperuricemia: Evidence from a Population-Based Prospective Cohort Study

**DOI:** 10.3390/ijerph20031728

**Published:** 2023-01-18

**Authors:** Zhibin Ma, Xiao Wang, Jia Zhang, Chao Yang, Hongmei Du, Feng Dou, Jianjian Li, Yini Zhao, Peiqin Quan, Xiaobin Hu

**Affiliations:** Department of Epidemiology and Health Statistics, School of Public Health, Lanzhou University, Lanzhou 730000, China

**Keywords:** chronic kidney disease, hyperuricemia, estimated glomerular filtration rate, cross lag panel model, cohort study

## Abstract

Background. Although several studies have examined the association between chronic kidney disease (CKD) and hyperuricemia (HUA), the direction of the association remains unclear. We aimed to investigate whether there was a bidirectional association between them. Methods. The present study was conducted in three analyses. Analysis I included 25,433 participants free of HUA at baseline to evaluate the associations between CKD and estimated glomerular filtration rate (eGFR) with incident HUA. Analysis II had 28,422 participants free of CKD at baseline to analyze the relationships between HUA and serum uric acid (sUA) with new-onset CKD. Cox proportional hazards regression models were applied to evaluate the association involved in Analysis I and II. Analysis III included 31,028 participants with complete data and further dissected the bidirectional association between sUA and eGFR using cross-lag models. Results. New-onset HUA and CKD were observed in the first round of the follow-up study among 1597 and 1212 participants, respectively. A significantly higher risk of HUA was observed in individuals with CKD compared to individuals without CKD (*HR* = 1.58, 95% *CI*: 1.28–1.95). The adjusted *HR*s (95% *CI*s) of HUA were 3.56 (2.50–5.05) for the participants in the group of eGFR less than 60 mL·min^−1^·1.73 m^−2^, 1.61 (1.42–1.83) for those in the group of eGFR between 60 and 90 mL·min^−1^·1.73 m^−2^, and 1.74 (1.42–2.14) for those in the group of eGFR more than 120 mL·min^−1^·1.73 m^−2^, compared with the group of eGFR between 90 and 120 mL·min^−1^·1.73 m^−2^. A higher risk of CKD was also observed in individuals with HUA compared to individuals without HUA (*HR* = 1.28, 95% *CI*: 1.12–1.47). Compared with the first quintile of sUA, the adjusted *HR* (95% *CI*) of CKD was 1.24 (1.01–1.51) for the participants in the fourth quantile. There was a bidirectional relationship between sUA and eGFR, with the path coefficients (*ρ*_1_ = −0.024, *p* < 0.001) from baseline eGFR to follow-up sUA and the path coefficients (*ρ*_2_ = −0.015, *p* = 0.002) from baseline sUA to follow-up eGFR. Conclusions. The present study indicated that CKD and HUA were closely associated, and there was a bidirectional relationship between sUA and eGFR.

## 1. Introduction

Chronic kidney disease [1] (CKD) is defined as a glomerular filtration rate (GFR) <60 mL·min^−1^·1.73 m^−2^ or the presence of one or more markers of renal injury for at least 3 months, which include albuminuria, abnormal urine sedimentation, and histological or structural abnormalities of the kidney. Patients with CKD are often complicated with cardiovascular disease (CVD) and progressing to end-stage kidney disease (ESKD). Therefore, CKD has a high risk of death and a heavy disease burden, which has become an urgent public health problem to be solved in China and even the world [2,3]. Over the past few decades, the global prevalence of CKD and the number of deaths caused by CKD have been increasing year by year. The overall global prevalence of CKD in 2017 was 9.10%. From 1990 to 2017, the global all-age CKD mortality rate increased 41.50%, and the ranking of causes of death also rose from 17th in 1990 to 12th [4,5]. Although the etiology of CKD is still unclear, it is significant to prevent the occurrence or delay of the progression of CKD by identifying the risk factors of CKD and intervening, which can reduce the risk of death and the disease burden of CKD [6].

Hyperuricemia (HUA) was not only found to predict the risk of cardiovascular disease [7] and metabolic syndrome [8]. Still, it was also considered one of the independent risk factors for the development of CKD, and the risk of CKD increased with the increase of serum uric acid (sUA) [9,10]. Accordingly, several cross-sectional studies have shown a significantly increased risk of HUA in people with CKD or reduced estimated glomerular filtration rate (eGFR) [11,12,13]. However, limited by their research design, the direction of their causal association cannot be determined, i.e., whether HUA is a cause or a consequence of CKD remains unclear, and longitudinal studies on them are still very limited.

Therefore, based on the above findings and the negative association between sUA and eGFR [14,15], we hypothesized that there is a bidirectional relationship between HUA and CKD, with HUA and CKD as follow-up outcomes in this cohort study, respectively. To clarify the associations between CKD and eGFR at the baseline and dynamic change of eGFR with incident HUA and the associations of HUA and sUA at the baseline and the dynamic change of sUA with new-onset CKD. Then, based on eGFR and sUA at the baseline and first-round follow-up, a cross-lagged panel model [16] was constructed to demonstrate the causal association and verify the bidirectional relationship between them.

## 2. Materials and Methods

### 2.1. Study Design and Participants

The participants were all drawn from the Jinchang cohort [17], an ongoing prospective cohort study in Jinchang City, Gansu Province, China, based on the biennial physical examination of all employees of Jinchuan Nonferrous Metals Company (JNMC). From June 2011 to December 2013, a total of 48,001 participants completed the cohort baseline survey, and 33,355 participants completed the first round of follow-up survey from January 2014 to December 2015, with a median follow-up time of 2.2 years.

This study was conducted in three analyses. Analysis I included 25,433 participants free of HUA at baseline to explore the associations between CKD, eGFR, and dynamic change of eGFR with incident HUA. Analysis II included 28,422 participants without CKD at baseline to analyze the relationships between HUA, sUA, and the dynamic change of sUA with new-onset CKD. Analysis III included 31,028 participants with complete data and further dissected the bidirectional association between sUA and eGFR. The specific inclusion and exclusion criteria and process of the study were shown in Figure 1.

### 2.2. Data Collection

The research data used in this study were derived from the Jinchang cohort baseline survey and the first round of follow-up surveys, including epidemiological questionnaires, physical examinations, and clinical biochemical examinations. Our research team designed the standardized and structured epidemiological questionnaires to collect basic sociodemographic information (age, gender, education, occupation, etc.), behavioral characteristics (smoking, drinking, exercise, etc.), and medical history of the participants. Uniformly trained interviewers conducted the questionnaire survey through one-on-one and face-to-face interviews. During the survey, it was ensured that the respondents clearly understood the content of the questionnaire, avoiding inducing questions, and cross-checking was conducted after completing the survey.

The physical examination and clinical biochemical examination were completed by the clinical staff of the Workers’ Hospital of the Jinchuan Company, including height, weight, blood pressure, and various clinical biochemical indexes. Height and weight were measured by a computerized body scale (SK-X80/TCS-160D-W/H, Sonka, China) when the participants took off their shoes and wore light clothes. Body mass index (BMI) was calculated as weight in kilograms divided by height in meters squared (kg·m^−2^). The blood pressure in a sitting position was measured by an electronic sphygmomanometer (BP750, AMpall, Seoul, Korea) three times continuously after at least 10 min of rest, and the average values were taken. Before venous blood collection, all participants were instructed to fast for at least 8 h. The clinical biochemical examination was detected by an automatic biochemical analyzer (Hitachi 7600-020, Kyoto, Japan), mainly including serum creatinine (Scr), sUA, total cholesterol (TC), fasting plasma glucose (FPG), triglyceride (TG), high-density lipoprotein cholesterol (HDL-C), and low-density lipoprotein cholesterol (LDL-C).

### 2.3. Study Outcomes and Related Definitions

CKD [1] was defined in this study as the presence of abnormal glomerular filtration rate (eGFR < 60 mL·min^−1^·1.73 m^−2^) or proteinuria (urine dipstick reading ≥ 1+), of which eGFR was estimated according to the Chronic Kidney Disease Epidemiology Collaboration equation (CKD-EPI), based on Scr, age, and gender [18]. Reduced eGFR was defined when it was less than 60 mL·min^−1^·1.73 m^−2^, otherwise normal. The eGFR estimated at baseline and the first round of follow-up were combined to define the type of dynamic changes in eGFR level, with the N-N group representing normal eGFR at baseline and follow-up, the N-R group representing normal eGFR at baseline and reduced eGFR at follow-up, the R-N representing reduced eGFR at baseline and normal eGFR at follow-up, and the R-R group representing reduced eGFR at both baseline and follow-up.

According to the internationally accepted epidemiological diagnostic criteria, HUA [19] in this study was defined as sUA >420 μmol/L in males and sUA >360 μmol/L in females, otherwise normal. The sUA measured at baseline and the first round of follow-up were combined to define the type of dynamic changes in sUA level, with the N-N group representing normal sUA at the baseline and the follow-up, the N-H group representing normal sUA at the baseline and hyperuricemia at the follow-up, the H-N representing hyperuricemia at the baseline and normal sUA at the follow-up, and the H-H group representing elevated sUA at both the baseline and the follow-up.

### 2.4. Covariates

Smokers were those who smoked at least one cigarette a day for more than 6 months, and non-smokers were those who never smoked or who smoked occasionally but did not meet the definition of a smoker. Ex-smokers were those who used to smoke but had not smoked for more than 6 months. Drinkers were those who drank liquor or other spirits, wine or other fruit wine, beer, and other alcohol at least once a week for more than 6 months, and non-drinkers were those who never drank or drank occasionally but did not meet the definition of drinkers. Ex-drinkers were those who used to drink but had not drunk for more than 6 months. Physical exercise was divided into three types: no, occasionally, and often exercise. Occasionally exercise was defined as exercise less than 3 times a week and exercise more than 30 min on average, and often exercise was considered as exercise at least 3 times a week for more than 30 min each time. Hypertension was defined as self-reported physician-diagnosed hypertension or definite clinical records of hypertension or blood pressure ≥140/90 mmHg (1 mm Hg = 0.133 kPa). Diabetes was defined as self-reported physician diagnosis of diabetes or definite clinical records of diabetes or fasting blood glucose ≥7.0 mmol/L.

### 2.5. Statistical Analysis

Participants’ baseline characteristics were presented as means ± standard deviation (SD) for continuous variables and numbers (percentages) for categorical variables. Comparison of continuous variables between groups using the Student’s *t*-test and chi-squared test for categorical variables. Hazard ratios (*HR*s) with 95% confidence intervals (95% *CI*s) were calculated to estimate the associations between CKD, eGFR at baseline and dynamic change of eGFR with incident HUA and the associations of HUA, and sUA at baseline and dynamic change of sUA with new-onset CKD by using Cox proportional hazards regression models, including Model 1 and Model 2. None of the covariates were adjusted for Model 1, covariates that were included in Model 2 were those that altered the hazard ratios for the effect of CKD and HUA by more than 5% in Analysis I and Analysis II, respectively, and all of the covariates were in the form of categorical variables. Finally, the covariates included in Model 2 of analyses I were age (<45 years, 45–64 years, ≥65 years), gender (male, female), BMI (<24.0 kg·m^−2^, 24.0–27.9 kg·m^−2^, ≥28 kg·m^−2^), education (junior high school or below, high school, junior college, bachelor’s degree or above), smoking status (non-smoker, smoker, ex-smoker), drinking status (non-drinker, drinker, ex-drinker), hypertension (no, yes), TC (≤4.20 mmol/L, 4.21–2.00 mmol/L, ≥5.01 mmol/L) at baseline. The covariates included in Model 2 of analyses II were age (<45 years, 45–64 years, ≥65 years), gender (male, female), BMI (<24.0 kg·m^−2^, 24.0–27.9 kg·m^−2^, ≥28 kg·m^−2^), smoking status (non-smoker, smoker, ex-smoker), drinking status (non-drinker, drinker, ex-drinker), diabetes (no, yes), hypertension (no, yes), TG (≤1.20 mmol/L, 1.21–2.00 mmol/L, ≥2.01 mmol/L) at baseline. Stratified analyses were performed according to age gender, BMI, smoking, and drinking status. Likelihood ratio tests were used to investigate interactions. Restricted cubic splines were used to investigate the possible nonlinear relationship between eGFR with incident HUA and sUA with new-onset CKD. Knots were placed at the 5th, 35th, 65th, and 95th percentiles, with the 50th percentiles set as a reference for sUA and eGFR, respectively.

The cross-lag panel model was used to analyze the bidirectional association between sUA and eGFR. The multiple regression equations of baseline and follow-up sUA, eGFR, and covariates were constructed before the analysis separately. The residuals of the above equations were taken and normalized by Z-transformation (mean = 0, SD = 1), and then the path coefficients were estimated by R package “lavaan”. The covariates included in Analysis III were follow-up time, age, gender (male, female), BMI, TC, TG, diabetes (no, yes), hypertension (no, yes), smoking status (non-smoker, smoker, ex-smoker), drinking status (non-drinker, drinker, ex-drinker), education (junior high school or below, high school, junior college, bachelor’s degree or above) when they were not the strata variables, among which the multi-categorical variables smoking, drinking and education were included in the model as dummy variables. The comparative fit index (*CFI*) and root-mean-square residual (*RMR*) were used to evaluate the model fitting, with *CFI* > 0.90 and *RMR* < 0.50 representing a good fit. Differences between path coefficients were tested by Fisher’s Z to identify the main causal sequential effect.

Several sensitivity analyses were also conducted. To avoid possible reverse causality, we performed sensitivity analyses after excluding participants who experienced outcomes during the first 2 years of follow-up in both Analysis I and II. To avoid possible bias in the glomerular filtration rate estimation equation, we estimated GFR using the modified Modification of Diet in Renal Disease (MDRD) equation and then assessed the bidirectional association between it and sUA. In addition, we reassessed the bidirectional relationship between sUA and eGFR after excluding patients with gout at baseline in Analysis III.

All statistical analyses were performed with SAS program, version 9.4 (SAS Institute Inc., Cary, NC, USA) and R software (R Foundation for Statistical Computing), version 4.2.1. All statistical tests were two-sided, and *p* < 0.05 was considered statistically significant.

## 3. Results

### 3.1. Baseline Characteristics

Among 25,433 participants eligible for Analysis I, participants with CKD at baseline had lower HDL-C and eGFR levels, and higher BMI, SBP, DBP, FPG, TC, TG, LDL-C, and sUA levels. They were more likely to be older, male, and have a higher proportion of diabetes and hypertension (Appendix A). Among 28,422 participants eligible for Analysis II, participants with HUA at baseline had lower HDL-C, and eGFR levels, and higher BMI, SBP, DBP, FPG, TC, TG, LDL-C, and sUA levels. They were more likely to be older, male, and have a higher proportion of hypertension (Appendix A). The baseline characteristics among the 31,028 individuals eligible for Analysis III were shown in Appendix A.

### 3.2. Analysis I: Associations between CKD, eGFR at Baseline, and Dynamic Changes of eGFR with New-Onset HUA

Among 25,433 participants, 1597 new-onset HUA were observed during 56,698.43 people’s years of follow-up, with an incidence density of 28.17/1000 person/year. As shown in Table 1 and Figure 2, a significantly higher risk of HUA was observed in individuals with CKD compared to individuals without CKD. The adjusted hazard ratio (*HR*) and 95% confidence interval (*CI*) were 1.58, 1.28–1.95, respectively. Results of stratified analyses showed that the association of CKD with risk of HUA was more evident among participants who were between 45 and 64 years old, female, ex-smoker and ex-drinker, or had a BMI < 24.0 kg·m^−2^, while no significant interaction was observed except for age (*p* for interaction = 0.010) and gender (*p* for interaction = 0.023; Figure 3).

Moreover, in addition to lower eGFR, an association between higher eGFR with new-onset HUA was also observed. Compared with group 3 of eGFR, the adjusted *HR*s (95% *CI*s) of HUA were 3.56 (2.50–5.05) for the individuals in the group1, 1.61 (1.42–1.83) for those in group 2, and 1.74 (1.42–2.14) for those in group 4 (Table 1). In addition, after adjusting for those same covariates as Model 2 in Table 1, except in subgroups aged between 45 and 64 years, older than 65 years, and white-collar workers, the results of restricted cubic spline showed that there were U-shaped relationships between eGFR with incident HUA (Appendix A).

Analyses were adjusted for age (<45 years, 45–64 years, ≥65 years), gender (male, female), BMI (<24.0 kg·m^−2^, 24.0–27.9 kg·m^−2^, ≥28 kg·m^−2^), education (junior high school or below, high school, junior college, bachelor’s degree or above), smoking status (non-smoker, smoker, ex-smoker), drinking status (non-drinker, drinker, ex-drinker), hypertension (no, yes), TC (≤4.20 mmol/L, 4.21–2.00 mmol/L, ≥5.01 mmol/L) at baseline when they were not the strata variables.

Importantly, a remarkably higher risk of HUA was observed in participants with a sustained reduction in eGFR. Compared with the group of N-N, the adjusted *HR* (95% *CI*) of HUA was 3.90 (2.63–5.77) for the participants in the group of R-R (Appendix A). The associations between CKD, eGFR, and dynamic change of eGFR with incident HUA remained unchanged after excluding participants who had outcomes during the first 2 years of follow-up (Appendix A).

### 3.3. Analysis II: Associations between HUA, sUA at Baseline, and Dynamic Changes of sUA with New-Onset CKD

Among 28,422 participants, 1212 new-onset CKD were observed during 63,574.32 person years of follow-up, with an incidence density of 19.06/1000 person years. As shown in Table 2 and Figure 2, a significantly higher risk of CKD was observed in individuals with HUA compared to individuals without HUA. The adjusted *HR* (95% *CI*) was 1.28 (1.12–1.47). Results of stratified analyses showed that the association of HUA with risk of CKD was more robust among participants who were over 65 years old, female, and either a non-smoker or ex-smoker, while no significant interaction was observed (all *p* for interaction > 0.05; Figure 4).

Analyses were adjusted for age (<45 years, 45–64 years, ≥65 years), gender (male, female), BMI (<24.0 kg·m^−2^, 24.0–27.9 kg·m^−2^, ≥28 kg·m^−2^), smoking status (non-smoker, smoker, ex-smoker), drinking status (non-drinker, drinker, ex-drinker), diabetes (no, yes), hypertension (no, yes), and TG (≤1.20 mmol/L, 1.21–2.00 mmol/L, ≥2.01 mmol/L) at baseline when they were not the strata variables.

Furthermore, the risk of CKD increased with the increase of sUA. Compared with the first quintile of sUA, the adjusted *HR* (95% *CI*) of CKD was 1.24 (1.01–1.51) for the individuals in the fourth quantile (*p*
_trend_ < 0.001, Table 2). Besides, after adjusting for those same covariates as Model 2 in Table 2, except in subgroups aged between 45 and 64 years and older than 65 years, the results of restricted cubic spline showed that there were positive linear dose–response relationships between sUA with incident CKD (Appendix A).

Moreover, a notably higher risk of CKD was observed in participants with persistently elevated sUA. Compared with the group of N-N, the adjusted *HR* (95% *CI*) of HUA was 1.83 (1.54–2.16) for the participants in the group of H-H (Appendix A). The relationships between HUA, sUA, and dynamic change of sUA with new-onset CKD persisted after excluding Individuals who had outcomes during the first two years of follow-up (Appendix A).

### 3.4. Analysis III: Cross-Lagged Panel Analysis between sUA and eGFR

A bidirectional relationship between sUA with eGFR was observed by using a cross-lagged model, with the path coefficients (*ρ*_1_ = −0.024, *p* < 0.001) from baseline eGFR to follow-up sUA and the path coefficients (*ρ*_2_ = −0.015, *p* < 0.01) from baseline sUA to follow-up eGFR, the difference between *ρ*_1_ and *ρ*_2_ was not statistically significant (Figure 5). Furthermore, these bidirectional relationships did not alter when the data were stratified by gender, and still existed among participants who were between 45 and 64 years old, front-line workers, and white-collar workers (Figure 5). When eGFR was estimated by the modified Modification of Diet in Renal Disease equation, the results remained similar among participants who were male, between 45 and 64 years old, over 65 years old, front-line workers, and white-collar workers (Appendix A). After excluding 195 participants with gout at the baseline, the above bidirectional relationships still exist (Appendix A). All models showed *RMR* values of 0.05 or less and *CFI* values of 0.90 or greater.

(A: Total population, B: male, C: female, D: age < 45 years old, E: age between 45 and 64 years old, F: age ≥ 65 years old, G: front-line workers, H: white-collar workers). Covariates included in the model were follow-up time, age, gender (male, female), BMI, TC, TG, diabetes (no, yes), hypertension (no, yes), smoking status (non-smoker, smoker, ex-smoker), drinking status (non-drinker, drinker, ex-drinker), and education (junior high school or below, high school, junior college, bachelor’s degree or above). These were not the strata variables, among which the multi-categorical variables smoking, drinking, and education were included in the model as dummy variables.

sUA, serum uric acid; eGFR, estimated glomerular filtration rate; *ρ*_1_, cross-lagged path coefficients from baseline eGFR to follow-up sUA; *ρ*_2_, cross-lagged path coefficients from baseline sUA to follow-up eGFR; *r* represented synchronous correlations; *β*_1_ and *β*_2_ represented tracking correlations; *R^2^*, variance explained; *RMR*, root mean-square residual; *CFI*, comparative fit index.

## 4. Discuss

The present study showed that CKD and HUA were closely associated with each other. There was a U-shaped relationship between eGFR with incident HUA and a positive linear dose–response relationship between sUA with incident CKD, with HUA and CKD as follow-up outcomes in a prospective cohort study, respectively. In addition, there was a bidirectional relationship between sUA with eGFR.

Uric acid [20] (UA) is the final product of purine metabolism in the human body, which is mainly excreted by the kidneys. When the kidney function is insufficient or declined, the excretion of UA is hindered, resulting in an increase in the level of sUA and the occurrence of HUA. A significantly higher risk of HUA was observed in individuals with CKD compared to individuals without CKD in the present study. Similarly, results from several cross-sectional studies [12,13,21] have also demonstrated an association between renal insufficiency and the risk of HUA, which increases with lower eGFR. However, these studies only analyzed the association between low eGFR (<60 mL·min^−1^·1.73 m^−2^) with the risk of HUA and did not further explore the possible association between high eGFR with HUA. Studies have shown that high eGFR can be used as a predictor of cardiovascular disease [22] and hypertension [23], while the association between high eGFR and HUA risk has rarely been reported. We found that the adjusted *HR*s (95% *CI*s) of HUA were 3.56 (2.50–5.05) for the participants in the group of eGFR less than 60 mL·min^−1^·1.73 m^−2^, 1.61 (1.42–1.83) for those in the group of eGFR between 60 and 90 mL·min^−1^·1.73 m^−2^, and 1.74 (1.42–2.14) for those in the group of eGFR more than 120 mL·min^−1^·1.73 m^−2^, compared with the group of eGFR between 90 and 120 mL·min^−1^·1.73 m^−2^, and that there were U-shaped relationships between eGFR with incident HUA. Studies have shown that GFR hyperfiltration not only leads to increased intra-glomerular pressure [24], but also higher levels of tubular markers, such as neutrophil gelatinase-associated lipocalin (NGAL) and kidney injury molecule-1 (KIM-1) in hyperfiltered individuals compared to those without hyperfiltration [25]. Besides, it can also lead to proteinuria and reduced renal function [26]. Therefore, the association between high eGFR and the risk of HUA in the present study may be due to the obstruction of UA excretion after a renal injury caused by GFR hyperfiltration. Prospective studies with larger samples are needed to validate these findings and to dissect the underlying mechanisms of these associations.

It was found that UA can lead to reduced eGFR and CKD probably through the formation of urate crystals that block the renal tubules, induce the proliferation of vascular smooth muscle cells and the reduction of endothelial NO, and activate the renin-angiotensin-aldosterone system [27,28]. These possible mechanisms provide a reasonable biological explanation for our results. We found that the adjusted *HR* (95% *CI*) was 1.28 (1.12–1.47) in individuals with HUA compared to individuals without HUA, and the adjusted *HR* (95% *CI*) of CKD was 1.24 (1.01–1.51) for the participants in the fourth quantile, compared with the first quintile of sUA. These findings are consistent with previous studies [29,30,31,32]. Importantly, repeated measurements of indicators and the study of associations between their dynamics with the disease can, to a certain extent, avoid the interference caused by potential reverse causality and make the results more accurate and reliable. For example, in a cohort study in Taiwan, China [33], sUA levels were measured repeatedly, significantly higher risk of CKD was observed among individuals whose sUA levels were above clinical cut-off values at baseline and follow-up, compared with individuals with sUA levels below clinical thresholds. Similarly, a notably higher risk of CKD was also observed among participants with persistently elevated sUA in the present cohort study. Additionally, a remarkably higher risk of HUA was observed among participants with a sustained reduction in eGFR as well as in the present study.

In addition, a bidirectional relationship between sUA with eGFR was observed in the present study by using a cross-lagged model, which has the advantage of controlling for the autoregressive effects of the variables and allows the direction of the main causal effect to be determined by comparing the differences between the cross-lagged path coefficients, provided that the time-series relationship is clear. It was found that eGFR decreases with age, and there may be gender differences in the association between sUA and eGFR [34,35]. Therefore, subgroup analyses in the cross-lagged model were performed by age and gender in our study. Likewise, these bidirectional relationships did not alter when the data were stratified by gender and still existed among participants who were between 45 and 64 years old. However, no association between sUA and eGFR was observed in the subgroup aged less than 45 years old. In the present study, the incidence of HUA was 3.85%, 7.91%, and 12.57% among those aged <45 years, 45–64 years, and ≥65 years, respectively, with the lowest incidence of HUA among those aged <45 years. Similarly, the incidence of CKD was 3.15%, 4.01%, and 11.22% in those aged <45 years, 45–64 years, and ≥65 years, respectively, with the lowest incidence of HUA in those aged <45 years. Because of the lower incidence of HUA and CKD in the age <45 years group, there were relatively few cases of elevated sUA and reduced eGFR in this population during the follow-up survey. We presume that this is the reason why a bidirectional association between sUA and eGFR was not observed in this population. Meanwhile, only a unidirectional significant association between baseline eGFR and follow-up sUA was observed in the subgroup aged more than 65 years old, which may be due to the relatively more severe eGFR decline in older participants [36,37], so we only observed that baseline eGFR significantly predicted follow-up sUA in this population (*ρ*_1_ = −0.058, *p* < 0.001), whereas baseline sUA has not been observed to predict substantially follow-up eGFR (*ρ*_2_ = −0.023, *p* = 0.073). Besides, the number of people aged ≥ 65 years in Analysis III (N = 3205) was smaller compared to those aged <45 years (N = 16,064) and 45–64 years (N = 11,759), which somewhat limited the statistically positive results. Therefore, further exploration of this result in a multicenter, larger-volume population is still needed, though whether their association differs in different age groups and the mechanism of the difference needs to be further explored.

To our knowledge, this cohort study with relatively large sample size is the first to investigate the bidirectional relationship between eGFR and sUA using cross-lagged panel analysis in a Chinese population. However, several limitations should be taken into account. First, this study used eGFR and albuminuria to jointly diagnose CKD, in which albuminuria was qualitatively diagnosed with urine dipstick, which was used to diagnose albuminuria in several epidemiological studies, rather than quantitative detection of urine protein, such as urine albumin/creatinine ratio (UACR). The study will be more comprehensive and reliable if the cross-lagged model can be used to further analyze the relationship between sUA and UACR as a supplement. Second, this study only included the data of the Jinchang cohort at baseline and the first round of follow-up surveys, with a relatively short follow-up period. Additionally, it was limited to the Jinchang cohort population, all participants were from JNMC, and most of them were front-line workers whose long-term exposure to industrial raw materials and heavy metals may have contributed to the development of HUA and CKD in this population, which limited the generalization of the study conclusion. Therefore, the results still need to be further verified in a multicenter, large-scale cohort study. As an observational study, although we adjusted for confounding factors as much as possible, it still could not avoid the interference of residual confounding. Finally, this study lacks the medication information and diagnostic information for the hematopoietic disorders of participants. Some patients with CKD or HUA may have been treated and taken the corresponding drugs, eGFR may be increased, and sUA may be decreased, which would be an underestimation of the association between them in this study. In addition, elevated sUA would underestimate this association due to participants suffering from hematopoietic disorders.

## 5. Conclusions

In conclusion, using a cross-lagged model, the present study elucidated the bidirectional association between CKD and HUA and further demonstrated this association between eGFR and sUA. Given the above research results, CKD and eGFR can be used as predictors of the risk of HUA, which has important guiding significance for the early prevention of HUA and the screening of high-risk groups. The management of HUA and sUA should also be taken into account in the prevention and control of CKD. Active treatment of HUA and control of sUA levels are conducive to the prevention of CKD and delay of the decline of eGFR.

## Figures and Tables

**Figure 1 ijerph-20-01728-f001:**
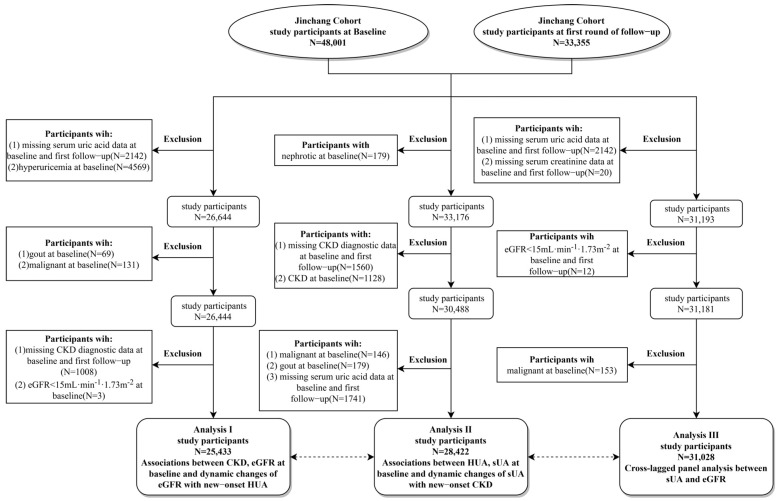
Inclusion and exclusion criteria and flow chart of study participants. sUA, serum uric acid; eGFR, estimated glomerular filtration rate; HUA, hyperuricemia; CKD, chronic kidney disease.

**Figure 2 ijerph-20-01728-f002:**
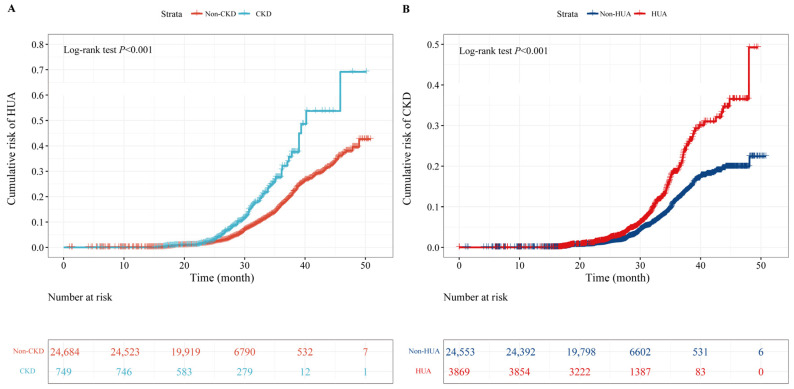
Cumulative incidence of HUA and CKD. ((**A**): cumulative incidence of HUA in participants with CKD compared with non-CKD, (**B**): cumulative incidence of CKD in participants with HUA compared with non-HUA). HUA, hyperuricemia; CKD, chronic kidney disease.

**Figure 3 ijerph-20-01728-f003:**
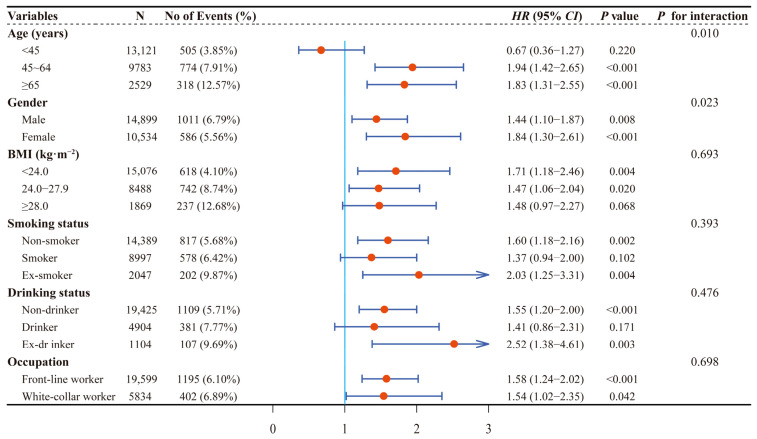
Stratified associations between CKD and new-onset HUA by age, gender, BMI, smoking status, drinking status, and occupation. CKD, chronic kidney disease; HUA, hyperuricemia; BMI, body mass index; HR, hazard ratio; CI, confidence interval.

**Figure 4 ijerph-20-01728-f004:**
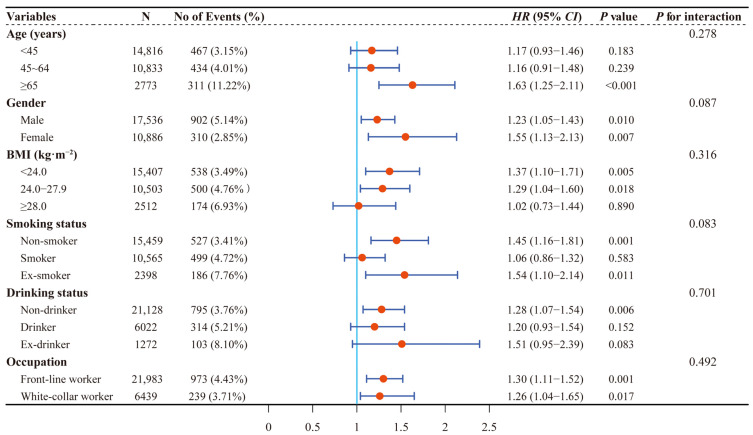
Stratified associations between HUA and new-onset CKD by age, gender, BMI, smoking status, drinking status, and occupation. HUA, hyperuricemia; CKD, chronic kidney disease; BMI, body mass index; HR, hazard ratio; CI, confidence interval.

**Figure 5 ijerph-20-01728-f005:**
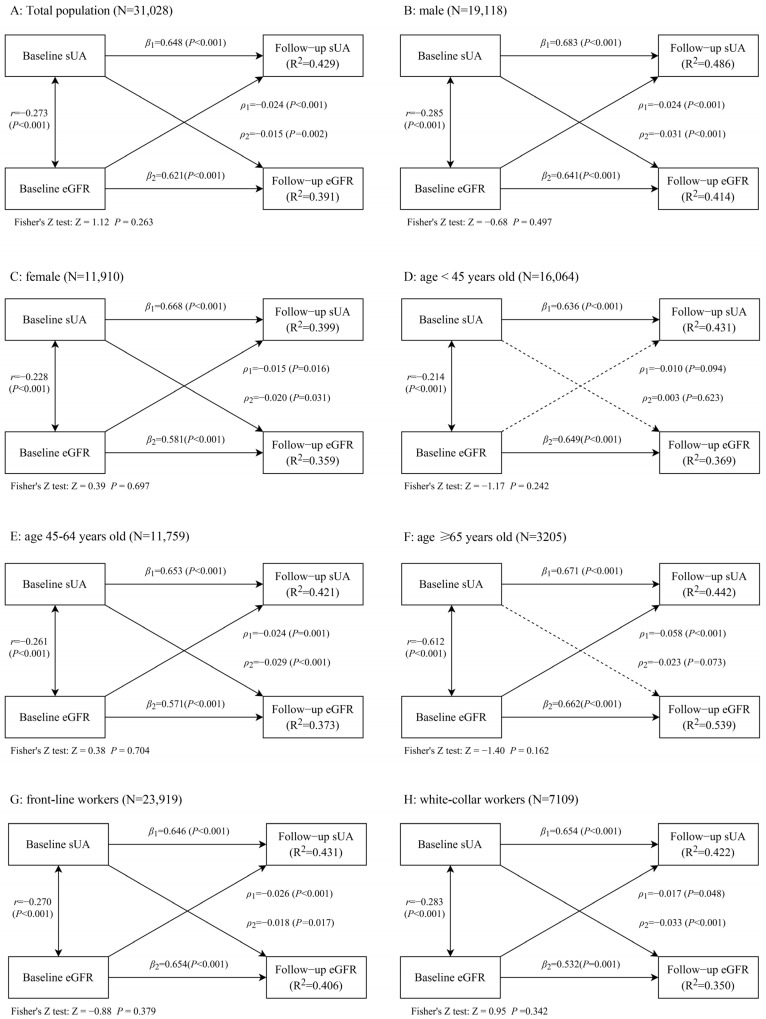
Sub-analyses of the cross-lagged model of the association between sUA with eGFR, stratified by gender, age, and occupation.

**Table 1 ijerph-20-01728-t001:** Associations between CKD and eGFR at baseline with new-onset HUA.

	N	No of Events (%)	Model 1		Model 2	
			*HR* (95% *CI*)	*p* Value	*HR* (95% *CI*)	*p* Value
CKD at Baseline						
Non-CKD	24,684	1502 (6.08%)	1.00		1.00	
CKD	749	95 (12.68%)	1.94 (1.58–2.39)	<0.001	1.58 (1.28–1.95)	<0.001
Groups of eGFR						
(mL·min^−1^·1.73 m^−2^)						
Group1 (<60)	135	35 (25.93%)	3.88 (2.77–5.43)	<0.001	3.56 (2.50–5.05)	<0.001
Group2 (60~<90)	3689	438 (11.87%)	1.72 (1.53–1.92)	<0.001	1.61 (1.42–1.83)	<0.001
Group3 (90~<120)	19,191	999 (5.21%)	1.00		1.00	
Group4 (≥120)	2418	125 (5.17%)	1.74 (1.44–2.10)	<0.001	1.74 (1.42–2.14)	<0.001

Model 1 was not adjusted for any covariates. Model 2 was adjusted for age (<45 years, 45–64 years, ≥65 years), gender (male, female), BMI (<24.0 kg·m^−2^, 24.0–27.9 kg·m^−2^, ≥28 kg·m^−2^), education (junior high school or below, high school, junior college, bachelor’s degree or above), smoking status (non-smoker, smoker, ex-smoker), drinking status (non-drinker, drinker, ex-drinker), hypertension (no, yes), TC (≤4.20 mmol/L, 4.21–2.00 mmol/L, ≥5.01 mmol/L) at baseline. CKD, chronic kidney disease; eGFR, estimated glomerular filtration rate; HUA, hyperuricemia; *HR*, hazard ratio; *CI*, confidence interval.

**Table 2 ijerph-20-01728-t002:** Associations between HUA and sUA at baseline with new-onset CKD.

	N	No of Events (%)	Model 1		Model 2	
			*HR* (95% *CI*)	*p* Value	*HR* (95% *CI*)	*p* Value
HUA at Baseline						
Non-HUA	24,553	933 (3.80%)	1.00		1.00	
HUA	3869	279 (7.21%)	1.70 (1.48–1.93)	<0.001	1.28 (1.12–1.47)	<0.001
sUA (μmol/L)						
Q1 (≤285)	7024	192 (2.73%)	1.00		1.00	
Q2 (266–318)	7087	260 (3.67%)	1.22 (1.01–1.47)	0.036	1.03 (0.85–1.25)	0.738
Q3 (319–374)	7091	309 (4.36%)	1.46 (1.22–1.74)	<0.001	1.04 (0.85–1.27)	0.705
Q4 (≥375)	7220	451 (6.25%)	2.02 (1.70–2.39)	<0.001	1.24 (1.01–1.51)	0.038
*p* _trend_			<0.001		<0.001	

Model 1 was not adjusted for any covariates. Model 2 was adjusted for age (<45 years, 45–64 years, ≥65 years), gender (male, female), BMI (<24.0 kg·m^−2^, 24.0–27.9 kg·m^−2^, ≥28 kg·m^−2^), smoking status (non-smoker, smoker, ex-smoker), drinking status (non-drinker, drinker, ex-drinker), diabetes (no, yes), hypertension (no, yes), TG (≤1.20 mmol/L, 1.21–2.00 mmol/L, ≥2.01 mmol/L) at baseline. Q1–Q4 referred to the quartiles of serum uric acid grouped according to the quartiles of the non-CKD participants, respectively. The median of each group was included in the regression model as a continuous variable to calculate the *p* value for the test of trend. HUA, hyperuricemia; sUA, serum uric acid; SBP, systolic blood pressure; CKD, chronic kidney disease; *HR*, hazard ratio; *CI*, confidence interval.

## Data Availability

The data underlying this article cannot be shared publicly due to data protection reasons. The data will be shared upon reasonable request to the corresponding author.

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
