# Peer review of "The Bidirectional Relationship between Chronic Kidney Disease and Hyperuricemia: Evidence from a Population-Based Prospective Cohort Study"

_ijerph, 2023, doi:10.3390/ijerph20031728_

Round 1

Reviewer 1 Report

The authors performed a prospective cohort study with 25433 participants included and concluded that CKD and HUA were closely associated to each other, there was a bidirectional relationship between sUA with eGFR. In general, the logic of the study is clear.

1.    The most significant disadvantage is that this study is a single center short-term follow-up research. All participants were workers in nonferrous metal factories and will long-term exposure to industrial raw materials and heavy metals lead to high uric acid and chronic CKD? It’s necessary to make subgroup analysis of front-line workers and white-collar workers and state the limitation of the research in the Discussion part.

2.     Why didn’t the authors include baseline variables of significant difference in the Cox proportional hazards regression models by convention?

3.     Please provide specific value of p value.

4.    No significant bidirectional relationship has been observed in the population aged<45 years and 65 years. Did this mean that the two-way relationship between the two disease was not stable? Please discuss the results further.

5.    Please provide accession number of the ethical approval document.

Author Response

Response to Reviewer 1 Comments

The authors performed a prospective cohort study with 25433 participants included and concluded that CKD and HUA were closely associated to each other, there was a bidirectional relationship between sUA with eGFR. In general, the logic of the study is clear.

 Point 1: The most significant disadvantage is that this study is a single center short-term follow-up research. All participants were workers in nonferrous metal factories and will long-term exposure to industrial raw materials and heavy metals lead to high uric acid and chronic CKD? It’s necessary to make subgroup analysis of front-line workers and white-collar workers and state the limitation of the research in the Discussion part.

Response 1: Thank you for your valuable comments, which have greatly assisted the manuscript revision and our future scientific work.

In Analysis I, we supplemented both the subgroup analysis of CKD and new-onset HUA (manuscript Figure ) and the restricted cubic spline of eGFR and new-onset HUA (Supplementary, Figure 1) with subgroup analyses stratified by occupation (frontline workers, white-collar workers). The results showed that the association between CKD with new-onset HUA persisted in frontline and white-collar workers, and the dose-response relationships between eGFR and new-onset HUA also existed.

Besides, in Analysis II, we supplemented both the subgroup analysis of HUA and incident CKD (manuscript Figure 4) and the restricted cubic spline of sUA and incident CKD (Supplementary, Figure 2). with subgroup analyses stratified by occupation (frontline workers, white-collar workers). The results showed that the association between HUA with incident CKD persisted in frontline and white-collar workers, and the dose-response relationships between sUA and incident CKD were also observed.

Moreover, in Analysis III, we supplemented the cross-lagged model with subgroup analyses stratified by occupation (frontline workers, white-collar workers). The results showed that the bidirectional relationship between sUA and eGFR was present in both frontline and white-collar workers (Supplementary, Figure 4). Finally, we discuss this limitation in the discussion section (Manuscript Discussion section, lines 501-504).

Point 2: Why didn’t the authors include baseline variables of significant difference in the Cox proportional hazards regression models by convention?

Response 2: Thank you for your valuable comments, which have provided critical assistance in the manuscript revision and our future scientific work.

We used the change in estimate in the Cox proportional hazards regression models to adjust for confounders, rather than including all variables with statistically significant differences in baseline characteristics in the model. The reasons for this are as follows.

The study showed that in situations in which the best decision (of whether or not to adjust) is not always obvious, the change-in-estimate (CIE) criterion tends to be superior[1]. Additionally, we included variables in Analysis I and Analysis II to adjust for confounders. The study showed that when the goal of variable selection is to select confounders, the change in estimate approach is often seen as preferable to methods based on statistical significance[2]. Importantly, in contrast with traditional predictive modeling, change in estimate strategies select covariates on the basis of how much their control changes exposure effect estimates; this observed change is presumed to measure confounding by the covariate. Many academics have recommended change-in-estimate rather than significance testing of the covariates [3-5].

[1]Mickey R M, Greenland S. The impact of confounder selection criteria on effect estimation[J]. American journal of epidemiology, 1989, 129(1): 125-137.[2]Greenland S, Pearce N. Statistical foundations for model-based adjustments[J]. Annual review of public health, 2015, 36: 89-108.[3]Breslow N E, Day N E, Schlesselman J J. Statistical methods in cancer research. Volume 1—The analysis of case-control studies[J]. Journal of Occupational and Environmental Medicine, 1982, 24(4): 255-257.                                                                            [4]Kleinbaum D G, Kupper L L, Morgenstern H. Epidemiologic research: principles and quantitative methods[M]. John Wiley & Sons, 1991.                  [5]Schlesselman J J. Case-control studies: design, conduct, analysis[M]. Oxford university press, 1982.

Point 3: Please provide specific value of p value.

Response 3: Thanks to your correction, we have uniformly verified the P-values of full text, with specific accounts for values of P>0.001, and P-values <0.001 were presented as P<0.001.

Point 4: No significant bidirectional relationship has been observed in the population aged<45 years and ≥ 65 years. Did this mean that the two-way relationship between the two disease was not stable? Please discuss the results further.

Response 4: Thank you for your valuable comments, which have provided critical assistance in the manuscript revision and our future scientific work.

In this study, the incidence of HUA was 3.85%, 7.91%, and 12.57% among those aged <45 years, 45-64 years, and ≥65 years, respectively, with the lowest incidence of HUA among those aged <45 years. Similarly, the incidence of CKD was 3.15%, 4.01%, and 11.22% in those aged <45 years, 45-64 years, and ≥65 years, respectively, with the lowest incidence of HUA in those aged <45 years. Because of the lower incidence of HUA and CKD in the age <45 years group, there were relatively few cases of elevated sUA and reduced eGFR in this population during the follow-up survey, and we presume that this is the reason why a bidirectional association between sUA and eGFR was not observed in this population.

As for the possible reasons for not observing a bidirectional association between sUA and eGFR in the age ≥ 65 years population, they are as follows. First, in general, eGFR decreases with age (physiologic age-related decline in eGFR) and is relatively more likely to be reduced in the age ≥ 65 population, so we only observed that baseline eGFR significantly predicted follow-up sUA in this population (ρ1=-0.058, P<0.001), whereas baseline sUA has not been observed to predict substantially follow-up eGFR (ρ2=-0.023, P=0.073).

Besides, the number of people aged ≥ 65 years in Analysis III (N=3205) was smaller compared to those aged <45 years (N=16,064) and 45-64 years (N=11,759), which somewhat limited the statistically positive results. Therefore, further exploration of this result in a multicenter, larger-volume population is still needed. We have added the above discussion to the discussion. Please kindly review (Manuscript discussion section, lines 465-482).

Point 5: Please provide accession number of the ethical approval document.

Response 5: Thanks again for your correction. The number of the ethical approval document is 2015-01, which has been added to the revised manuscript (line 533).

Reviewer 2 Report

In this study, Zhibin Ma et al. investigated the relationship between Chronic kidney disease and hyperuricemia in a prospective cohort study. They tackled an important unanswered question to determine the directionality of their causal association.

The tables, figures, and data all seem to be in order. The figures specifically are of adequate quality of resolution.

I would recommend a minor English check. The sentences are all clear, but a few have some grammatical errors.

Other than that, I do not have any remarks regarding the design of the study or the presentation of data. 

Author Response

Response to Reviewer 2 Comments

In this study, Zhibin Ma et al. investigated the relationship between Chronic kidney disease and hyperuricemia in a prospective cohort study. They tackled an important unanswered question to determine the directionality of their causal association.

The tables, figures, and data all seem to be in order. The figures specifically are of adequate quality of resolution.

I would recommend a minor English check. The sentences are all clear, but a few have some grammatical errors.

Other than that, I do not have any remarks regarding the design of the study or the presentation of data.

Point 1: I would recommend a minor English check. The sentences are all clear, but a few have some grammatical errors.

Response 1: Thank you for your valuable comments, which have provided critical assistance in the manuscript revision and our future scientific work.

In response to your comments, we carefully reviewed the language issues word by word, and asked native English-speaking researchers in this research field to revise and inspect the manuscript, based on which we have made some changes, which involve numerous statements and therefore do not indicate the line numbers one by one, so please review it again.

Reviewer 3 Report

Ma and colleagues present a very sophisticated analysis of the relationship of CKD and hyperuricemia.

The manuscript is well written, the results are clearly presented and the study design is appropriate. The conclusions are backed up by the results and the outstanding part are the statistical analyses performed with a data base that is absolutely sufficient for the performed analyses.

Although the study presents novel findings that are relevant for professionals in these fields, one minor point is missing:

1. As hyperuricemia is hugely influenced by chronic diseases resulting in higher UA levels due to e.g. cell proliferation (like gout and hematopoietic disorders and malignancies), does the data include the number of patients suffering from these conditions? Information on these diseases should be reported, as they can influence the UA levels per se and in disease attacks.

2. Directly associated with the 1. point, how many patients were under UA lowering medication?

Author Response

Response to Reviewer 3 Comments

Ma and colleagues present a very sophisticated analysis of the relationship of CKD and hyperuricemia.

The manuscript is well written, the results are clearly presented and the study design is appropriate. The conclusions are backed up by the results and the outstanding part are the statistical analyses performed with a data base that is absolutely sufficient for the performed analyses.

Although the study presents novel findings that are relevant for professionals in these fields, one minor point is missing:

Point 1: As hyperuricemia is hugely influenced by chronic diseases resulting in higher UA levels due to e.g. cell proliferation (like gout and hematopoietic disorders and malignancies), does the data include the number of patients suffering from these conditions? Information on these diseases should be reported, as they can influence the UA levels per se and in disease attacks.

Response 1: Thank you for your valuable comments, which have greatly assisted the manuscript revision and our future scientific work.

Our study was divided into three parts, of which both Analysis I and Analysis II excluded patients diagnosed with gout and malignancy at baseline (Figure 1, line 82 of the manuscript), which is not shown in the manuscript text but is clearly explained in the inclusion and exclusion criteria and flow chart of study participants While analysis 3 excluded only patients with malignancy but not patients with gout at baseline, as a complement, we performed sensitivity analysis after excluding 195 participants with gout at baseline in analysis III, and the results showed that the bivariate association between sUA and eGFR still existed (Supplementary Figure. 4; lines231-233, lines 370-371 of the manuscript).

Due to a large number of participants in the Jinchang cohort at the time of the baseline survey and the constraints of conditions and funding at that time, a definitive diagnosis of hematopoietic disorders was not made, so patients with hematopoietic disorders were not excluded from the Analysis I, II, and III, which we have added to the limitations of the revised manuscript study (Manuscript discussion section, lines 508-513). We will take this into full consideration in future studies to avoid bias due to such diseases as much as possible.

Point 2: Directly associated with the 1. point, how many patients were under UA lowering medication?

Response 2: Once again, thank you for your valuable comments, which have provided vital assistance in the revision of this manuscript and future scientific work.

Similar to the previous response, due to a large number of participants in the Jinchang cohort at the time of the baseline survey and the constraints of conditions and funding at that time, we did not collect information on the medications taken by the study participant, which we have explained in the limitations of the discussion section of the original manuscript (line 508-513).

Round 2

Reviewer 1 Report

Line numbers in the rebuttal letter were confused. Revisions were acceptable.